# Position: Federated Learning is a Lens towards a Democratized Future for the Scaling Law Era

**Harry H. Jiang** [1]   **Baris Askin** [1]   **Gauri Joshi** [1]   **Carlee Joe-Wong** [1]

## Abstract

Machine learning (ML) systems have grown significantly in size and popularity over recent years. However, the data and computation power supply chains which have helped fuel this growth have not been built without controversy. In particular, some of the data used to train these models may have been used without permission, while the growing appetite for compute power in model training increasingly incentivizes consolidation of access to larger players. As some stakeholders, such as data owners and everyday consumers of the Internet, have felt left behind by the emerging ML ecosystem, we seek to use federated learning paradigm as a model and motivation to develop a more democratized future for the ML community: one that is more decentralized, cooperative, and accountable. This position paper argues that the original proposition of federated learning as a framework enabling cooperation, privacy, and decentralization is still relevant today, even after the emergence of large foundation model- and scaling law-driven ML research, and that FL can inspire alternative ML ecosystems which alleviate and avoid the current frictions of large ML systems.

## 1. Introduction

As ML (machine learning) becomes a technology which is implemented at increasingly large scale, the formation of new supply chains for the data and computation resources required to train ever-growing models has proven to be controversial. For example, several companies building large models have faced lawsuits based on their methods for acquiring training data (Stempel, 2023; Brittain, 2023), and the ever-growing need for more compute power has raised concerns about risky investments in datacenter re-

sources (Raitano, 2025; Dang, 2024; Weise & Tan, 2025). In the midst of a fraught relationship between ML developers and other stakeholders in the production of ML models, we place a spotlight on the subfield of federated learning as a case study in re-emphasizing *user-centric* values and incentives in the ML field. In this paper, we take federated learning as a lens through which a truly democratized ML future, that is, one less consolidated and more accountable to all stakeholders large or small, can be imagined and built.

**Federated learning** (FL) (Daly et al., 2024; McMahan et al., 2017) is a distributed training framework in which multiple clients, e.g., company locations or mobile devices, each with their own local dataset, aim to collectively train a model on this local data (Figure 1). Instead of simply sending this local data to a central server for model training, the federated learning clients themselves train local copies of a model. These are occasionally aggregated throughout the training process, typically at a central server, so that each client's local model can benefit from other clients' data. FL thus allows clients a measure of control over access to their data, which may have privacy benefits, and naturally makes use of computation power and datasets that may be distributed across multiple clients, e.g., consumer data housed on individual mobile devices.

As data consent, privacy, and hardware access are points of friction that the ML field is encountering, we argue that **FL can be a starting point for designing an alternative ML ecosystem that is more decentralized, cooperative, and accountable.** We give an overview of FL and its current applications in Section 2 before making our main points:

**- FL alleviates concerns on proper consent and compensation to access training data** (Section 3). We first discuss current controversy on early models' training on publicly available data without data owners' consent, as well as the challenges of ensuring that data owners will be fairly compensated when their data is used in model training. We then argue that FL frameworks can facilitate an alternative, open market for training data in which data owners and model builders can more fairly negotiate for access to training data.

**- Interest in FL from data privacy-regulated industries points towards FL's ability to safeguard private data**

[1]Carnegie Mellon University, Pittsburgh, Pennsylvania, USA. Correspondence to: Harry H. Jiang <hhj@andrew.cmu.edu>.

*Proceedings of the 43rd International Conference on Machine Learning*, Seoul, South Korea. PMLR 306, 2026. Copyright 2026 by the author(s).

(Section 4). By keeping training data at the clients, FL provides natural avenues to protect the privacy of this data. While privacy in FL is well-studied, this further points to FL's ability to facilitate a more user-centric ML ecosystem.

**- FL can help alleviate consolidation in the current market for computing resources** (Section 5). We first discuss the potential systemic risks posed by the seemingly ever-growing appetite for compute resources to train large models. We then argue that, by naturally distributing training across physically distinct clients, FL can reduce direct computing demand from model builders and alleviate demand pressures on the market for compute hardware.

Using FL as a motivating example, we also encourage ML researchers to envision and design system-level innovations in which we can help resolve conflicts the field is facing.

We also present counterpoints to our position. First, that FL is not inherently conducive to building alternative futures (Section 6): the application space for FL is currently limited to only a few areas, many developments from the FL area have contributed to the centralized approach, and certain applications still depend on powerful institutions which concentrate data and resources. Second, given the popularity of generative AI application, the ML field has achieved a new status quo for which the inherent challenges of FL make it ill-equipped to change the ecosystem at large (Section 7). We present our rebuttals to these arguments and conclude by reiterating our position in Section 8.

## 2. Background

In this section, we first give an overview of federated learning and its current applications, and then discuss the emergence of foundation models that underlie much of the current ML ecosystem. We also present similar work discussing the intersection of economics and ML.

### 2.1. Federated Learning

Federated learning (FL) is a form of distributed machine learning, which is distinguished by its property of having disjoint training data that does not leave its clients, motivated in part by principles of data privacy (McMahan et al., 2017). These clients then aim to train models on their collective local datasets, despite the restriction that data remain at the client. Often used with a hub-and-spoke topology (or typically referred to as server-client), FL can also be characterized as a type of distributed ML technique in a setting featuring at least one of the following distinct challenges: (I) heterogeneous client data (Jhunjhunwala et al., 2023), (II) heterogeneous client devices (Wang et al., 2020b), (III) communication constraints (Konečný et al., 2018), (IV) privacy constraints (McMahan et al., 2018), and (V) unreliable clients (Dutta et al., 2021).

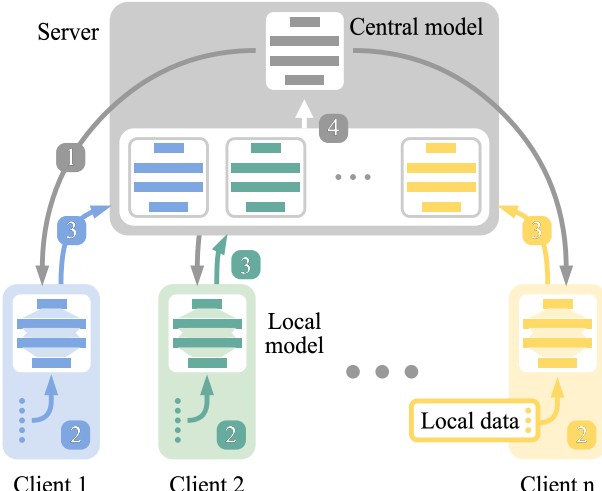

*Figure 1.* The most rudimentary form of FL, FedAvg (McMahan et al., 2017), consists of a framework where a central server holds the most up-to-date model parameters and coordinates *rounds* in which a subset of clients (1) receive the centralized parameters, (2) updates them using their local data for a number of epochs, and (3) returns them to the server which (4) updates the central model with the average of returned parameters from participating clients. Though FL techniques have gotten significantly more sophisticated, they typically use a similar server-client topology.

Federated learning as a paradigm was proposed by McMahan et al. (2017), in which the FedAvg algorithm (Figure 1) was presented originally as a way to train a model using a loose *federation* of mobile devices without them needing to share their private data. FedAvg, and most FL algorithms, rely on a central aggregation server to receive and combine model updates from multiple clients, thus allowing them to effectively make use of each other's data during training. Subsequent developments have identified the key challenges in the federated setting, as listed above, as well as further key topics in (VI) personalization (Li et al., 2021), (VII) fairness (Cho et al., 2024), and (XIII) adversarial robustness (Wang et al., 2020a).

### 2.2. FL in Practice

Applications of FL have coalesced into two large categories: **cross-device FL**, the setting with mass, typically consumer-owned, edge devices tackled in the original proposition of FedAvg; and **cross-silo FL**, the setting which coordinates across fewer *institutional clients* with more computing power and data available to each client. In cross-device FL, a model builder typically acts as the central aggregation server, and client devices might be smartphones or sensors that collect local data about their users or environments. In cross-silo FL, the model builder would also act as a central aggregation server. Clients may consist of separate companies, e.g., different hospital systems, or different divisions within a company, e.g., divisions from North America, Eu-

rope, Asia, etc. may all serve as separate FL clients.

In industry, interest in FL has largely concentrated on cross-silo FL. Common areas of implementation are in healthcare (where regulations such as GDPR and HIPAA[1] constrain patient data within organizations or even physical premises), finance (also under consumer data regulation), manufacturing (for protecting intellectual property), and government (Kuo et al., 2025). Both industry and academia have also consistently expressed interest in pre-training and fine-tuning large language models through FL (Daly et al., 2024; Raje et al., 2025)

### 2.3. Emergence of Foundation Models

The early 2020s marked the emergence and entrenchment of an ecosystem for ML models based on large, pretrained models (also known as "foundation" or "base" models) and downstream finetuning for applications. This shift in focus for the ML community is often viewed as a result of three factors converging: the introduction of Transformers (Vaswani et al., 2017) in natural language processing (NLP), the maturation of transfer learning into in-context learning (Radford et al., 2019), and the postulation of neural scaling laws (Kaplan et al., 2020; Henighan et al., 2020). We can identify the culmination of these factors in the release of GPT-3 by Brown et al. (2020) and the subsequent surge in popular interest for using language models (OpenAI, 2025).

The last of these three factors is especially impactful to the ML community as it proposes an extrapolate future model performance almost purely as a function of its input resources: compute budget, dataset size, and model parameters. A simple roadmap for increasing capability emerged in the *scaling law era*: increasing each input resource yields better results. Though scaling laws in ML, like previously proposed curve effects such as Moore's Law, were never meant to be extrapolated indefinitely, its immediate validity has turned the laws into an *incentive* for developers in recent years. Coupled with the commercial success of LLM-based products such as ChatGPT, this resource acquisition strategy also became a business imperative.

Entrenching the pretrained model ecosystem is the emergence of general-purpose generative models. The potential of pre-trained languages models such as GPT-1 (Radford et al., 2018) has been massively bolstered by the proposition of in-context learning by Radford et al. (2019) and Brown et al. (2020). This spawned a belief that large language models (LLMs) are generalizable to any text-based task, and that pursuing large scale will only improve its generalizability. The development of multi-modal NLP-supervised models such as CLIP (Radford et al., 2021) then led to large

generative models in other modalities such as Stable Diffusion for the image domain (Rombach et al., 2022). These large general-purpose models became known as *foundation models*, a term coined by the Stanford Institute for Human-Centered AI (2021) to identify the broad applicability for these models and the potential of any specific task to be achieved by the adapting one such model.

Surrounding these foundation models, a segment of the ML research community and much industry interest in ML has concentrated on pursuing two broad objectives: the training of ever-larger and comprehensive foundation models, and the adaptation thereof to specific tasks.

### 2.4. Related Work

As the intersection of economics and ML is an emerging and fast-moving field, we highlight recent position papers in the ML community on the economics of ML which discuss the data market and its societal effects (Santy et al., 2025; Jia et al., 2025; Vincent et al., 2025), as well as technical work on modeling and optimizing data pricing and markets (Xu et al., 2025; Ai et al., 2025; Zhang et al., 2026; Clinton et al., 2026; Tang et al., 2026). Also, Einav & Rosenfeld (2025) explores ML models under competition. Regarding the social welfare aspect of ML on the economy, Hazra et al. (2025) promotes analysis of labor impacts of AI systems, while the intersection of economics and ML regulation is explored in Qiu et al. (2026).

Adjacent to our work, McCoy et al. (2025) also challenges the current incentive of pursuing scaling and presents an alternative objective to incentivize ML research rooted in sustainability. Similar to our premise, Baek et al. (2025) promotes a research direction within ML, adaptive sensing, as a solution to concerns with scaling law-based incentives for ML research.

## 3. Data Strategies and Incentives

One of the most critical input resources for producing ML models is training data. Since the entrenchment of the scaling incentive within the ML field, companies in the area have been in need of increasing orders of magnitude in data, but early practices in data collection by model builders have set a foundation for an uneven data market. We posit that federated learning systems can motivate the design of an alternative data market where buyers and producers of data can participate fairly.

### 3.1. The Scraping Economy

The ML community has historically relied on using publicly available datasets to evaluate new ideas and train new models. As datasets increase in size and as more ML research and development is conducted by private companies,

---

[1]General Data Protection Regulation (EU, 2016) and Health Insurance Portability and Accountability Act (USA, 1996)

this practice began attracting controversy, with even early large language models such as Google's BERT attracting ire from the Authors Guild for using BooksCorpus, a University of Toronto and MIT-curated dataset of free e-books (Lea, 2016). BooksCorpus was similarly used by GPT-1, and for GPT-2, OpenAI scraped pages linked from a set of filtered Reddit posts. In either case, permission to use the data for training was not requested from their owners.

While early LLMs used highly-curated, platform-specific datasets, they are not large enough to produce a pre-trained model at larger scales. GPT-3 in 2020, for instance, has augmented the training set of its predecessor with scraped web data from Common Crawl (originally used by Google in (Raffel et al., 2020)), as well as Wikipedia and two since-deleted datasets named Books1 and Book2 (Brown et al., 2020). In the image domain, the LAION (Schuhmann et al., 2022) and DataComp (Gadre et al., 2023) projects have released massive datasets from scraped data. This data collection was at first uncompensated from the data provider or owner's perspective, and many from those stakeholders consider it a blatant attempt of profiting from their resources (Henderson et al., 2023; Jiang et al., 2023).

Given that data has become a critical input commodity for the production of these large models, it is reasonable to question whether data owners are receiving fair compensation for their data. These *data owners* include large players such as media conglomerates (e.g., Disney, Associated Press, etc.), Web platforms (e.g. Reddit, Meta, Adobe, etc.), and trade associations (e.g., Authors Guild, which previously brought a lawsuit against the Google Books tool on behalf of its authors[2]). They also include smaller entities, in particular small publishers, individual authors and artists, independent websites, and in growing frequency, consumers whose personal data is of interest to model trainers.

On the other side of the market, *data buyers* primarily consist of model builders (e.g., OpenAI, Google, Meta, Adobe, etc.). We argue that the dominance of large players on both the buyer and, arguably, the seller sides of the market for pre-training data resembles that of a bilateral oligopoly. Under an oligopoly, pricing is dominated by a few sellers; a bilateral oligopoly means that large players control price determination from both the supply and demand sides. While smaller sellers do often own data that can be valuable to model buyers, small players like individual consumers often do not have access to mechanisms, much less resources, to negotiate with large model builders for access to their data.

We further argue that the initial practice of uncompensated scraping for training data on the part of model builders puts the market for data at an unfair starting point. This behavior has led large players in the media space to either sue for

restitution or negotiate for deals (or both), while individual creators of intellectual property seek consent, credit, and compensation included as part of fair value for their data (Kyi et al., 2025). The bind that data owners and creators face is that pursuing adequate restitution occurs at a cost for the sellers (e.g., in time, opportunity, legal representation), while letting the issue go has zero cost, but that the outcomes of these decisions influences future pricing in the data market at large. This is highly problematic for the sellers, i.e., the data owners, who have to spend money to discover upper bounds of acceptable value. It especially hurts small data owners, as large data owners or brokers, which form the supply side of the emerging bilateral oligopoly, may not be transparent about pricing or interested in spending the amount needed for optimal price discovery. Indeed, they may even effectively drive down the open market price of data through parallel asset or equity agreements which only sellers of a certain size are equipped to achieve. Examples of such moves include the Reddit-Google licensing deal, which never released a confirmed price tag, though estimated to be valued at $60 million (Tong et al., 2024), and the Disney-OpenAI licensing and equity agreement for $1 billion in investment into the latter (The Walt Disney Company, 2025).

### 3.2. FL as an Alternative Data Market

We argue that federated learning, being a paradigm based on client cooperation, can be a model for an alternative market that can help ensure data owners are fairly compensated for the use of their data to train models. Since models in federated learning are trained at the clients, i.e., where the data is generated, data providers must actively consent to and participate in training tasks. This allows for more choice for suppliers, i.e., the data owners, to choose which models they wish to help train and thus incentivizes model builders to cooperate with data owners and negotiate fairly. In an open market where both buyers and sellers have clearly defined negotiation pathways and an easy means of refusal, transparent price discovery can more easily occur, leading to more optimal mutual benefit than the current regime and a bulwark from uneven negotiation power and collusive behavior found in typical oligopolies.

FL can further change characteristics of the data market as the traded commodity in an FL-mediated market actually becomes *model updates* instead of the data itself: participating in FL training means that a client sends model updates to the aggregation server, but the server still cannot otherwise access clients' local training datasets. One benefit to this training structure is that model updates, being an *intermediate* product in the training process, are less susceptible to unfair pricing that depends on the perceived quality of a client's raw data, as the trainer does not have an immediate way to estimate contribution towards a model's performance

---

[2]*Authors Guild v. Google*, 804 F.3d 202 (2nd Cir. 2015)

like they would with data. The contribution of a client can be measured through a model update which evaluates the client as a whole, including data quality and computational quality. This is especially true for newcomer clients, whose updates are *a priori* essentially fungible with others at mean price. For repeat clients, this property acts as a regularizer of price, where their model updates will be priced fairly compared to similar peer clients. Ultimately, this protection against price discrimination serves as a stronger incentive for data owners to participate than in the murky market which exists now.

Interactions between players in an FL-mediated marketplace are complex, as they involve multiple players that could each have a range of incentives. Thus, incentive design itself is also an open area of research in FL which can bolster the applicability of FL as a relief for tensions in the data market. Examples include auction models for client participation and price-setting methods by the central server based on data quality and computation capacity estimates (Siew et al., 2025). Vulnerabilities in FL such as data and model poisoning, free-riding, and backdoors can be alleviated with proper incentivization mechanisms, and free-riding in particular touches on incentive challenges, e.g., requiring a certain level of "useful" client participation in order to either receive payments or have access to the trained FL model (Karimireddy et al., 2022). Much recent work in FL has additionally developed training frameworks that address poisoning (Kang et al., 2020) and backdoor vulnerabilities (Pene et al., 2024), which may be incorporated into incentivization frameworks.

## 4. Privacy Benefits

As mentioned in Section 3.1, increasing amounts of personal data stored on Web platforms are being used to train models, not to mention data scraped without knowledge. In addition to the issue of consent in data use, this situation also raises *privacy* concerns, especially as generative models are increasingly susceptible to data leakage from memorization (Carlini et al., 2019; Nasr et al., 2025; Carlini et al., 2023). Thus, malicious users with access to a trained generative model could recreate another user's private training data. While this concern is well-known in the ML community, the best practice(s) for safeguarding the privacy on training data are still actively evolving.

### 4.1. Federated learning and privacy

Federated learning's initial value proposition from 2017 included potential privacy benefits due to keeping data local to the clients (McMahan et al., 2017), as discussed in Section 2.1. Nine years after FL's introduction, these benefits remain relevant: the adoption of FL has largely occurred in the form of cross-silo applications in industries with per-

sonal data regulations such as healthcare and finance (e.g., under GDPR and HIPAA), or settings where intellectual property is sensitive, like manufacturing (Kuo et al., 2025). Privacy in FL, however, has proven to be more complicated than simply keeping raw training data local. Such data locality is not a standalone privacy guarantee, since shared model updates can still leak sensitive information. Reconstruction attacks can recover representative or even exact training samples from gradients/updates (Zhu et al., 2019). Despite that, FL is superior to centralized training, which provides no privacy guarantees due to the data gathering process in the first place, and FL is uniquely suited to integration with such privacy-preserving mechanisms in order to prove end-to-end privacy guarantees.

To address these privacy risks, in practice, FL is commonly paired with cryptographic protections such as secure aggregation, ensuring the server learns only aggregated updates, not any individual contribution. In some deployments, homomorphic encryption is used to compute aggregates over encrypted updates that do not reveal training data information (Near & Darais, 2024). For formal privacy guarantees, federated training can incorporate user-level differential privacy (DP) (Xu et al., 2023). Using cryptography protects updates in training, while DP bounds what an attacker can infer even from the final or intermediate models. Moreover, recent FL privacy work emphasizes auditability: participants should be able to verify what was computed and what guarantees were enforced (Daly et al., 2024).Other work such as (Jin et al., 2023) has examined how to navigate the tradeoffs of computational overhead with privacy guarantees in FL settings.

### 4.2. Privacy and training large models

As model training shifts towards including more privacy-sensitive, e.g., consumer, data, privacy will likely prove to be an ongoing concern. Considering the concerns about data use and privacy, we invite ML systems designers to consider the best practices adopted from these application areas and the affordances of FL. For example, FL can work with other federated frameworks such as Mastodon or Bluesky, which are composed of various decentralized servers named "instances" which can willingly federate with the social networks; users belong to a specific instance and their data is stored within it (Zignani et al., 2018). Such a structure naturally lends itself to a FL-like training framework, where each participating server can act as a client. A model could then train on data served on participating servers while maintaining consent in participation and user privacy. Within a single large organization, FL-based protocols (e.g., Eichner et al. (2025) and DiLoCo as proposed by Douillard et al. (2024) from Google) can help silo data across various locales, improving privacy and security of stored data and maintaining compliance of data onshoring regulations like

GDPR, or even distribute and train personalized models for end-users (e.g., Ji et al. (2025)).

# 5. Computing Resources

The second critical ingredient for training large ML models is access to computing power. The market for computing hardware is brushing up against supply constraints, which, similar to the data market, is seeing concentrated purchasing power. We posit that this development is reducing fair access to computers and building systemic risk in the computing sector, that is, risk that the entire sector sees widespread failure, due to the increased dedication of resources towards a few ML vendors, the lack of immediate returns from said vendors, and the debt accrued in investing in such projects. We see federated learning as an alternative which can reduce concentration and re-incentivize fair access to computing resources for businesses and consumers.

## 5.1. Systemic Risk in Computing Markets

**Demand for computing hardware.** As models grow, demand for computing power from ML applications scales up to match. Paired with the explosion of consumer applications using the largest models such as ChatGPT and Claude, hardware for ML is entering a constrained market regime where supply of computing power is limited by manufacturing capacity, especially that of memory (Leswing, 2026). As demand growth surpasses supply elasticity, prices for components sharply grow and skew purchasing power towards larger buyers.

The AI/ML industry's hunger for memory has cascading effects throughout the computing world as well. Consumer device production is directly affected, with prices for consumer products increasing due to constrained memory supply (Clark, 2026) and Micron, a leading memory manufacturer, even exiting the consumer market (Micron, 2025). The crunch in availability of high-quality compute in edge devices, alongside increasing data collection by large model builders and growing frequency of ML inference tasks, incentivizes platform providers to structure computing services to be hosted server-side.

**Consolidation incentives.** As a result of the above trends, the current ML economy is producing overwhelming incentives to consolidate computing power, and purchasing power, across the information technology (IT) domain into the hands of a few players. A corollary to that is that companies are adopting aggressive strategies to concentrate hardware and access thereof, such as securing supply through agreements with chipmakers and inking cloud service deals with popular platforms (Forgash & Ghosh, 2025). On the other hand, smaller tech firms, both within and outside the AI industry, face pressure to scale up centralized service infrastructure while contending with earlier movers' locked-in supply deals.

While increased demand and monetary investment sounds attractive for the IT sector, this concentration of demand poses significant risks. On the business end, the AI platform and hyperscaler industries are becoming leveraged[3], arguably to a dangerous degree, while lenders and investors face building concentration risk. One might recall that OpenAI is still losing money, but promises 12-figure revenue by the end of the decade while raising 11-figure capital investments (The Economist, 2025). Should revenue fall short of these forecasts, investors and datacenter operators could be at significant financial risk (Weise & Tan, 2025). The collapse of a large AI firm can have dire downstream effects: hyperscalers can lose significant portions of their revenue, impacting operational liquidity[4], which in turn would affect the availability of their services to other customers. All of these firms will see reduced revenue, which will hamper their ability to pay back debt; simultaneously, the leverage used on the firms will put investors' credit at risk. As a result, portions of the financial sector will face illiquidity, with ripple effects throughout the global economy. Thus, an over-leveraged oligopoly in computing power that is incentivized to dedicate growing proportions of the global computing capacity towards not-yet-profitable applications from only a few clients is a powerful gamble. It not only deprives computation capacity from other uses, but also introduces system instability into the sector at large should investments fail to punctually deliver returns to investors.

## 5.2. FL as a Comprehensive Market Solution

We argue that alleviating these concentration risks in the market for computing hardware may require system-level alternatives to deliver better access to computing for all users and businesses. Federated learning is in a natural position to provide such an alternative. At the basic level, FL as a decentralized paradigm undoes some of the effects of concentration by shifting training computation to smaller parties (i.e., clients), thus reducing the overall demand volume directly from model builders. While large buyers and sellers may still exist in an FL market, this training structure disincentivizes many of the negatives to an oligopolistic regime, such as concentrated purchasing power and systemic exposure to the risk of revenue loss from large customers that collapse, either from external circumstances or lack of service from the hyperscaler.

First, FL promotes higher resource utilization by allowing "loose" computing power at the clients to more easily contribute to model training, improving market efficiency and potentially reducing demand for dedicated ML training hard-

---

[3]i.e., having borrowed funds for investing

[4]i.e., cash on hand used for providing goods and services

ware. Especially in the cross-device setting, model builders can make use of consumer edge devices, which often sit idle, to supplement the computation supply in the cloud. This architecture may also incentivize moving model inference away from a centralized service model to the edge and promote customer-centric developments such as model personalization. It can further help relieve pressure on the non-dedicated ML segment of the computing market as well, as large firms have to then consider the impacts of availability of high-quality edge devices.

Second, the fact that the traded commodity in an FL-based market is the model update (Section 3.2) can reduce power imbalances in the market for compute, as model updates are in fact a proxy for *both data and computing power*: they require both to be computed. The price of model updates is thus backed by the basic value and cost of computing power, which is stable and fungible, in addition to the regularized value of data discussed in Section 3, which makes for a more tractable, physically backed valuation of a client's utility to model developers (since the developers now also depend on the clients to provide the bulk of the computing resources for training). Moreover, a market of potential clients carrying both data and computing power allows each to contribute to FL projects more freely. We can draw analogues to the current healthcare IT market, already an FL application area, where software vendors depend on data from customers of their IT services to train ML models (Fabbri, 2019). The utility of customer-stored data to an IT provider gives health institutions more power in negotiations and agency in selecting and switching vendors (consider that a customer leaving would mean the vendor losing access to a source of fresh data), and the latter have to adapt to maximize mutual utility in return.

Similarly, continuing the analogy with social media platforms in Section 4.2, user migration (Jeong et al., 2024), deletion (Minaei et al., 2022), and abstention behavior informs data participation behavior under a less centralized regime. Data for social media, even in a federated framework between institutions, still come from individual users, which have the choice of server through which they participate in the social network. Specifically, we expect users which are potential data sources to change contributing institutions within an FL network if they disagree with the institution's data usage policy, if not refuse participation outright. In such a scenario, the existence of choice between various silos on the data supplier (i.e. user) side incentivizes FL-contributing institutions to respect users' wishes more carefully and reduces the power of platforms to coerce users to provide data for training on the former's terms.

Federated learning's effects on the compute market may be particularly profound in sectors with data regulations where data may have to be processed within disparate institutions,

such as healthcare and finance. Each of these institutions, particularly when acting as a FL client, is thus incentivized to procure high-performance computing power itself. Under a federated framework of computationally well-endowed institutions acting as clients, model builders and vendors can more easily distribute both training and inference, thus reducing ML platforms' concentration of computing demand.

## Alternative Views

## 6. Counterpoint: FL Does Not Help Suggest Alternative Futures

Our first counterpoint to our argument that FL can be a path towards a more user-centric ML ecosystem is that FL will not help realize such fundamental changes.

### 6.1. FL Has Limited Applicability

After 9 years of active research, the ML research community and industry have generally mapped out the most realistic applications of FL, and there has been ample time to observe industry interest and substantiated applications in the field. FL has been adopted nearly exclusively in the context of sectoral limitations, whether on data privacy (e.g., healthcare, finance), security (e.g., manufacturing, defense), or communication cost (e.g., automotive, Internet of Things (IoT)). It would be difficult to promote adoption of a method which centers these constraints in environments where they do not exist. Moreover, the implementation challenges of building FL frameworks at scale, such as client participation, selection, synchronization, and latency, make FL unenticing for many model builders (Daly et al., 2024).

Furthermore, some of the developments stemming from FL that have seen wider applicability are ostensibly no longer within the FL realm in terms of the primary principles and challenges through which FL is defined. For example, while inspired by FedAvg, DiLoCo sees applicability in distributed training beyond federated settings; Charles et al. (2025) directly compare DiLoCo with data-parallel distributed ML in a similar setting. The area of federated analytics leaves ML altogether and is more appropriately called a data science tool (Ramage & Mazzocchi, 2020). Thus, FL itself has limited applicability.

### 6.2. FL Does Not Break Out of Platform Economics

**Federated learning and market power.** From a *macro*economic and critical lens, FL does not tackle the structural conditions that underlie some of the issues we observe in the ML ecosystem. Ultimately, FL still requires a central orchestrating party, which keeps it under the influence of platform economics, i.e., the set of economic and social effects resulting from Web platforms. In such a

market, platforms have coercive powers over participants through the manipulation of a network's technical functionalities. For example, in an FL setting, a model builder can use obscure opt-out mechanisms for participants with opaque algorithmic pricing to drive down prices. This is especially true if the model builder also owns a Web platform, as is often the case in practice. Also, FL does not explicitly disincentivize the scraping behaviour described in Section 3.1; this can only be achieved with a negotiated or regulated consensus between the supply and demand side, or through enforced regulation.

Aside from the potential actions of the model builder acting as an aggregation server in FL, FL may not effectively mitigate data oligopolies either. The utilities that clients receive from participating in a FL network are not independent from each other; as more clients (and thus more data and compute) contribute to training a FL model, the model's performance will improve. Thus, clients may have an incentive to band together and train the same model. Such effects maybe particularly pronounced since large companies like media conglomerates control large amounts of data that can make or break a model's success. Individuals may then need to follow their lead in order to participate in training effective models.

**Economic externalities.** As an Internet-age technology, ML has relied on Web-enabled crowdwork to process the data needed for training; for example, early foundational dataset ImageNet relied on Amazon Mechanical Turk (MTurk) for human labeling (Deng et al., 2009). A consequence of the need for ML training to scale up is the decreasing "quality" of training data. Starting with GPT-3, pre-trained language models have needed reinforcement learning from human feedback (RLHF) for alignment prior to release (Brown et al., 2020). To obtain such data, companies rely on large-scale paid outsourcing firms (e.g., the former Scale AI), which use on-demand workers to complete tasks. This practice pose a large externality on labor markets, especially in developing economies that are frequently outsourcing destinations for crowdwork; this externality can be controversial, with mixed views about their effects on various labor markets around the world (Heeks, 2017). In any case, a training paradigm like FL would not address this major effect of platform economics.

### 6.3. Rebuttal: FL Is a Reasonable Starting Point for Computer Systems Design

As part of an emerging and fast growing industry, ML practitioners should expect the arrival of new AI regulations across the world. We argue that FL is highly compatible with regulation, given its prioritization for data privacy in its design, and other accountability mechanisms such as auditing (see Section 4). Thus, it is a natural part of a proac-

tive stance promoting research in accommodating future regulation. For example, ongoing research in privacy and cryptographic guarantees in FL will further help the ML field adapt in a transition to a more regulated AI sector, where application constraints may resemble those already present in areas where FL is being adopted. Also, centralized aggregators may implement FL and purchase model updates if that is the only legitimate way to access data for items sensitive to users such as personal data and browsing behavior. They could pay users for raw data, but this also introduces significant communication overhead due to the need to transfer this data to a central location and retains security vulnerabilities from communicating sensitive information, which may not be allowed under privacy regulation.

Moreover, to extend Section 2.1, some areas of research in FL can improve its applicability and utility in tackling market concentration. Peer-to-peer implementations of FL, e.g. Wink & Nochta (2021), can alleviate the outsize power of a central coordinator in conventional FL. Also, areas which are thought to pose a challenge to deployment at scale, such as communication overhead and slower convergence, may turn out to be less of a concern to deployers (Kuo et al., 2025), and more readily traded off for market-wide efficiencies like higher utilization of available computing resources, especially as model training is entering a regime which is bottlenecked by computing demands.

While the ML industry is unquestionably subject to many structural forces beyond the technology itself, we believe it is reasonable to motivate computer researchers to explore computer systems-based solutions which can facilitate changes to the economy. In that regard, we believe that FL is a complement to efforts in other fields, such as economic policy, in more equitably distributing the benefits of advancements in ML/AI.

## 7. Counterpoint: FL Cannot Meaningfully Help Reform the ML Ecosystem

Our second counterpoint to this paper is that, even if FL can conceptually address issues of market power in data and computing markets, it would not be effective in practice.

### 7.1. Larger Models Are Too Economically Enticing

While the most relevant scaling laws of ML models still hold true and heavier ML-powered applications gain in popularity, it is difficult for the industry to turn away from the incentive of building larger, more resource-intensive models. Training a large foundation model has fewest operational challenges when centrally organized and computed, and is currently the preferable engineering and business solution.

Many of the business decisions surrounding AI/ML investment are informed by product-level characteristics, not

mode of production; for example, the future profit-making capacity of ChatGPT is more important to an investor than the way a GPT model is trained. Also, under the profit maximization assumption, in a high barrier-to-entry market such as the computing sector, individual investors and firms are incentivized to become large players and entrench themselves within an oligopoly. In the AI industry, this will reflect in the form of firms preferring concentrated control of resources and vertical integration of supply chains; FL systems that distribute control of resources and have inherently weaker vertical integration will not be as attractive to investment as the *status quo*.

### 7.2. Privacy Was Never That Important

Since most new ML applications are based on large base models, which are already hard to deploy on the edge, inference tasks often require client data to be sent to a centralized organization. This therefore negates the initial proposition of FL being a regime where data does not leave a client and trained locally: if inference already requires data transfer, then privacy risks on training data are comparatively moot. Given the track record of privacy violations by many Web platforms to which people have entrusted their data (e.g., Cadwalladr & Graham-Harrison (2018); Isaac et al. (2017)), and the relatively light penalties for these incidents (Somerville, 2016), it is difficult to convince consumer and regulators to keep privacy front-of-mind and hold platforms accountable. Furthermore, memorization is more than ever a problem for generative models (see Section 4), and it is possible that large generative models of the kind developed today turn out to be fundamentally incompatible with the privacy objectives of FL.

### 7.3. Rebuttal: FL Can Help Prepare for What's Coming

We believe that the systemic risk discussed in Section 5.2 is necessary to tackle regardless of current resistance. FL provides a technical off-ramp for the growing market concentration and systemic risks we observe in the current AI market. Moreover, just as integrated circuit density tapered off Moore's Law predictions in the 2010s, current neural scaling laws will eventually plateau. Whether that occurs sooner or later, we can be sure that the incentives of today will change; a plateau in scaling can be expected to redirect focus of the ML industry towards finding efficiencies and developing lighter, more focused models. There is a wide surface of exploration in terms of making ML applications more agile, less computationally or data intensive, and reliant on producing new pretrained models (e.g., efficient tuning (Raje et al., 2025; Cho et al., 2024; Woisetschläger et al., 2024; Yang et al., 2025; Sun et al., 2024), and partial edge offloading (Liang et al., 2026; Zhang et al., 2024)) FL unlocks scattered devices running below full utilization, and it can allow effective development of smaller ML applica-

tions making use of edge devices.

On privacy, reiterating our point in Section 6.3, FL is highly compatible with regulatory mechanisms, and this is an important consideration as governments begin to regulate ML applications for consumer protection. Just as prior cases of negligence towards data privacy and security led to legislation such as GDPR, the ML field must be ready for the imposition of new constraints.

## 8. Conclusion and Call to Action

In this work, we identify areas of tension of the ML industry from an economics lens, namely the consent and compensation for collecting training data, privacy concerns, and unsustainability of computing power concentration. We post that FL is not only still relevant in the current day, but a topic for ML researchers to understand and make use of as a potential starting point for imagining alternative ML ecosystems in which the developments in our field addresses these tensions and provide more benefit to all stakeholders.

Also, we believe that certain open questions in FL privacy are also relevant to the ML field at large, and we agree with recent work arguing that some topics in FL are underserved, as there is currently a misalignment between academia and FL in industry (Kuo et al., 2025). As such, we propose the following research objectives to the ML community:

- Research in ML model design, applications, and infrastructure that are conscious of system-wide effects, such as concentration of computing power and capital.
- ML research which does not follow the incentives formed by scaling laws, i.e. increasing data and computation needs.
- ML research which considers the consent and privacy of individuals.
- Research on participation incentives in ML systems.
- Research on cross-silo FL and intermediate regimes between cross-device and cross-silo.
- FL and systems research on achieving higher utilization of devices.

Ultimately, FL is only one of many motivating examples of alternative system-level frameworks; our larger objective is to encourage the ML community to have higher-level considerations in mind in their exploration and design of ML systems and deployment models. We focus our paper on FL due to the interest that FL has generated in the ML community in recent years, and its potential to incentivize active client participation in training machine learning models. We highlight other FL-adjacent examples include FlexOlmo (Shi et al., 2025), a federated mixture-of-experts paradigm, and the area of federated analytics (e.g., Chadha et al. (2024)), as areas in which FL-like ideas can flourish and enable more democratized ML systems.

## Acknowledgements

This work was supported by the National Science Foundation under grant CNS-2106891, CCF-2045694, CNS-2112471, and CPS-2111751, as well as the Office of Naval Research under grant N00014-23-1-2149. Any opinions, findings, and conclusions or recommendations expressed in this material are those of the authors and do not necessarily reflect the views of the National Science Foundation or Office of Naval Research.

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
