# OpenReview forum: "Position: Federated Learning is a Lens towards a Democratized Future for the Scaling Law Era"
_ICML.cc/2026/Position_Paper_Track — ICML 2026 Position Paper Track regular_

### Official Review · Reviewer_sYD3 · 2026-03-09

**Significance:** 2
**Argument Clarity:** 3
**Rating:** 3
**Confidence:** 4

**Questions:**

Please see the weakness. I think the main question would be why FL is the solution but not other privacy-preserving methods. Although the paper state that they are not saying FL is the only solution, but I think as a position paper the author need to be confirm at the main position point rather than weaken it.

**Alternative Views Section:**

Yes

**Compliance With Llm Reviewing Policy A Conservative:**

Affirmed.

**Discussion Potential:**

3

**Final Justification:**

The rebuttal partially solved my questions but some parts remain unclear for me. I still feel that claiming FL as a solution to the problems mentioned in the paper as a position is not an enough contribution in an ICML 2026 paper. I keep my score and will leave the justification to AC for this point.

**Paper Summary:**

This paper argues that federated learning can be viewed as a framework for building a more decentralized and accountable machine learning ecosystem in the era of foundation models. The authors discuss current concerns around data ownership, privacy, and the increasing concentration of computing resources, and suggest that FL’s paradigm of training on distributed data while keeping it local may offer an alternative to centralized model development. The paper also discusses practical limitations and rebuttal towards counterarguments, while discussing how FL could contribute to more balanced data and compute markets. The paper discuss whether FL can inspire more decentralized ML system designs. As a position paper, the position is clear that it calls the research community to pay more attention to FL for more data and higher data privacy guarantee.

**Position:**

Yes

**Position In Title:**

Yes

**Related Work:**

2

**Strengths And Weaknesses:**

The paper addresses an important topic, discussing federated learning as a potential lens for rethinking the current centralized ML ecosystem. The position is generally clearly stated and the paper is well organized, with a logical structure that includes background, arguments, counterpoints, and rebuttals. The weakness regarding the position itself is that the democratized ML + FL is already analyzed by a series of previous FL works (but of course, mainly focus on small models). This limits the significance of the proposed position in this paper.

The topic is relevant to the ICML community, which is very relevant to the compute resources in large-scale AI systems. The paper is also likely to stimulate discussion, since it connects a FL with broader economic and societal questions. The limitation of this part is, the paper does not provide clear enough logic that how will FL solve the problem of foundation models. The application of FL in foundation models is limited by the limited computational resource and communication bandwidth. The paper mention this point, but this seems preventing FL from solve the problem claimed by the position. So how can FL act is a lens towards a democratized future for the scaling law era? The authors not clearly support this position in the paper.

The paper does a good job outlining the problems it aims to address. The concerns are reasonable and well motivated, and the paper presents them in a clear and accessible way. However, the main weakness is that the argument for why FL specifically serves as the solution to these problems is not fully developed. While FL is presented as a promising direction, the paper does not sufficiently explain why FL is uniquely suited to address these issues compared to other possible approaches. As a result, the position itself is reasonable and interesting, but the link between the identified problems and the proposed solution remains somewhat weak, which makes the overall argument less convincing.

**Support:**

2

---

> ### Author Rebuttal · Authors · 2026-03-31
>
> We thank the reviewer for engaging with our position and providing us with constructive comments. We will incorporate our responses, shown below, into the revised manuscript.
>
> Why FL for privacy. We believe FL is complementary to other areas of privacy- and security-aware ML, such as homomorphic encryption and differential privacy, and that advancements in privacy further augment the viability and benefits of FL. These may, of course, come with additional computational overhead, and the resulting performance tradeoffs and their larger economic effects would be an interesting area of study.
>
> Regarding the reviewer’s comments on affirming FL in our position, we agree that we can improve the communication of our point that FL is an area in which we unequivocally promote research while remarking that FL is complementary to the adjacent topics in ML privacy, security, and decentralization. Please also see our response to Reviewer VwJp, point W3.
>
> Novelty relative to existing FL work. We agree that prior works have examined FL’s ability to democratize training for small models–indeed, we argue that FL, through its reliance on client participation in model training, naturally lends itself to a more democratized machine learning ecosystem. However, we are not aware of any prior work that explicitly connects these effects on machine learning training to broader economic effects, e.g., concentration of power in the ML ecosystem or compute resource markets. Our call to action (see our response to Reviewer Jmdi, point Q1) aims to encourage more research into such a broader lens on FL and the larger economic effects that its deployment (or the deployment of similar alternative training frameworks) can have on the machine learning ecosystem.
>
> Computational and communication resources. As this topic is similar to other reviewers’ comments on computational capabilities, we direct the reviewer towards our rebuttal for Reviewer VwJp, points W2 and W4.
>
> We would also like to highlight our analysis in Section 5, which identifies systemic risks to the economy at large  from the current form of the ML industry. We believe this discussion is timely and necessary for the ML community as we command an outsize influence on the economy. Given the lack of economics analysis in much of ML literature, we believe that our work in making this point explicit constitutes a contribution to the discourse within the ML community.

---

> > ### Author Rebuttal · Reviewer_sYD3 · 2026-04-03
> >
> > Thank you for the rebuttal, I still have the following questions regarding the rebuttal:
> >
> > Repetitive framing: “FL is complementary” is stated multiple times without adding new substance.
> >
> > Weak novelty claim: Saying “we are not aware of prior work” is not a strong contribution—needs clearer differentiation.
> >
> > Lack of self-contained response: Frequent cross-references make this hard to follow on its own.

---

### Official Review · Reviewer_Jmdi · 2026-03-11

**Significance:** 3
**Argument Clarity:** 2
**Rating:** 4
**Confidence:** 3

**Questions:**

Please address the weakness section.

The authors seem to provide various advocations regarding FL. Could you clarify what the primary claim or call to action is?

**Alternative Views Section:**

Yes

**Compliance With Llm Reviewing Policy A Conservative:**

Affirmed.

**Discussion Potential:**

3

**Final Justification:**

I thank the authors for the second round of rebuttal, which partially addressed my original concerns. The additional calls to action and the references to PEFT/LoRA, split FL, and black-box tuning, lighter model training as concrete FL training directions for edge-constrained devices are good additions. The clarification on how individual users retain meaningful agency through institutional choice in cross-silo FL systems is also helpful. In general, I agree that the position is timely and raises important questions worth discussing in the community. I increased my score to 4.

**Paper Summary:**

This paper argues that federated learning (FL) is still relevant in the era of foundation models and scaling laws. The authors position that FL can serve as a starting point for developing a more decentralized, cooperative, and accountable ML ecosystem for all stakeholders.
The paper makes three main claims: 1) FL can help facilitate an open and fairer data market via consent-based data participation; 2) FL provides natural protection to data privacy, which benefits stakeholders; 3) FL can help alleviate systemic risks arising from the concentration of computing resources. The paper also proposes two counterpoints: 1) FL has limited capacity and applicability; 2) FL cannot break platform economics, as it still requires a central orchestrating party.

**Position:**

Yes

**Position In Title:**

Yes

**Related Work:**

3

**Strengths And Weaknesses:**

Strengths:

1) The motivation of the position is clear and timely. Unfair data market, privacy concerns, and systemic risk in computing markets are relevant real-world issues in the era of foundation models. The paper provides clear evidence to illustrate these points.

2) The authors admit the applicable limitations of FL that are difficult to implement or change the current ML systems.

Weakness:

1) The final call to action (encourage the community to develop more democratized ML systems) is too general, and the whole discussion on FL in the era of scaling raw is vague. FL is a general paradigm that has many variants. The paper simply refers to the most general terminology without concretely discussing the potential real implementation and how it is applied in the current era. For instance, due to the large model size and limited memory and computing power on edge devices, standard cross-device FL becomes tricky. Such specific discussion and call to action are missing throughout the whole paper.

2) The paper does not properly discuss the communication bottleneck, which is the main difference between the centralized distributed training and FL.

3) The paper does not explain the fact in detail that the computing power between the data provider and the model training company is typically not at the same scale, which makes standard FL tricky to implement.

4) The discussion and explanation on the data market mechanism are not fully clear and are not well-defined.

5) While the paper contends that FL is only one of many motivating examples of alternative ML frameworks, the authors do not provide specific alternatives and do not explain why FL is a more promising lens compared to the others.

**Support:**

2

---

> ### Author Rebuttal · Authors · 2026-03-31
>
> We would like to thank the reviewer for the useful comments, which will help us refine our manuscript and have aided us in identifying areas in which we can improve the presentation of our arguments. We detail our responses to the specific weaknesses raised below, and we will add these points to the manuscript:
>
> Q1: We agree that our call for action is broad, and we will articulate specific goals for the research community in the revised manuscript:
> Research in ML model design, applications, and infrastructure that are conscious of system-wide effects, such as concentration of computing power and capital
> ML research which does not follow the incentives formed by scaling laws, i.e. increasing data and computation needs.
> ML research which considers the consent and privacy of individuals, for which we believe FL is one of the most constructive and also most compatible with other methods
> Research on incentives for participating in ML systems
> Cross-silo FL research and research on intermediate regimes between cross-device and cross-silo
> FL research on achieving higher utilization of devices
> All of these goals fall under our argument that FL can serve as a lens towards a more democratized ML future. Thus, our call to action is a call to the research community to support and conduct more research towards addressing these challenges.
>
> Q2: Communication overhead is indeed a challenge in FL, and several FL research papers have aimed to address this challenge. Recent surveys of industry practitioners in [1], however, have indicated that communication challenges are not considered particularly strong by existing practitioners in cross-silo applications. Thus, we do not believe that communication overhead necessarily precludes adoption of FL or similar frameworks.
>
> [1] K. Kuo, C. Yadav, V. Smith. “Research in Collaborative Learning Does Not
> Serve Cross-Silo Federated Learning in Practice”. 2025. https://arxiv.org/abs/2510.12595
>
> Q3: As this question is similar to other reviewers’ comments on computational capabilities, we direct the reviewer towards our rebuttal for Reviewer VwJp, points W2 and W4.
>
> Q4: We apologize for any confusion. In this paper, we argue that the interactions between players in the FL space (in particular data creators, data platforms, and model builders) can be modeled as a marketplace, in which model builders seek to obtain data from creators or platforms in order to train their models. Due to the non-technical nature of a position paper, we omitted several details of these potential market mechanisms. We acknowledge, however, that these interactions are complex, as they involve multiple players that could each have a range of incentives. Thus, we do not claim that our marketplace model is definitive. Indeed, one of our goals in writing this position paper was to spur more research into these marketplace interactions and how they would be affected by wider-scale FL deployments.
>
> Q5: We focus our position paper on FL due to the enormous interest that FL has generated in the machine learning community in recent years, and its natural tendency to incentivize active client participation in training machine learning models. Other learning frameworks, however, may achieve some characteristics of a democratized ML ecosystem. For example, model builders could directly pay data creators for access to noisy or synthetic yet realistic data, but the training itself could be done on a platform owned by the model builder. Such a framework might alleviate some data privacy concerns but would not address concerns about a centralized market for computing resources. Alternatively, some data platforms might act as federated learning clients, training local models on their own data and sending them to model builders, while others might directly sell data to model builders. We do not advocate for one of these frameworks over another (indeed, one of our goals in writing this paper was to encourage researchers to propose new frameworks that address some of these concerns). We will clarify some of these alternative frameworks in the revised manuscript.

---

> > ### Author Rebuttal · Reviewer_Jmdi · 2026-04-01
> >
> > I thank the authors for the detailed rebuttal. In general, I agree that the argument has good discussion potential.
> >
> > My feeling is that the arguments and evidence provided in the paper and rebuttal mostly support/favour cross-silo FL involving institutions, computing centers, and companies with sufficient computational resources.
> > The central argument that FL can help build a democratized future where individual data owners and everyday consumers of the Internet can benefit in the foundation model era still seems to have comparatively little concrete support. The connection between this motivating vision and the practical scope of the proposed solutions remains a bit unclear to me.
> >
> > In particular, the main bottleneck, limited edge device memory and computing power, seems still impractical to address.
> > I encourage the authors to include at least one concrete call for action specifically targeting this gap. For example,
> > a research proposal addressing how resource-constrained individuals could meaningfully contribute to FL systems where models have billions of parameters.

---

### Official Review · Reviewer_Kzjs · 2026-03-12

**Significance:** 3
**Argument Clarity:** 2
**Rating:** 4
**Confidence:** 4

**Questions:**

1. Today's foundational models routinely feature hundreds of billions of parameters. How feasible is it to overcome limitations in memory , computational power, and communication bandwidth when implementing such models across edge devices in cross-device scenarios?
2. How does the system address model quality degradation when confronted with highly non-iid data distributions, low-quality data, or even malicious poisoning attacks?
3. Ordinary users possess devices with limited computational power and highly fragmented data. What economic incentive motivates centralized aggregators to purchase these inefficient, high-communication-cost “model updates”? How can these users be incentivized to participate in the FL market?

**Alternative Views Section:**

Yes

**Compliance With Llm Reviewing Policy A Conservative:**

Affirmed.

**Discussion Potential:**

2

**Final Justification:**

The original intention of this article is very good. The author addressed some of my concerns, so I decided to raise the score.

**Paper Summary:**

This position paper argues that in an era where machine learning pursues large models, federated learning (FL) holds promise to steer the field toward a more democratized, decentralized, and responsible future. Addressing core challenges such as legal disputes over unpaid data scraping, privacy leakage risks, and financial vulnerabilities stemming from highly concentrated computing power markets, the authors demonstrate how FL can mitigate these crises by fostering fairer data exchange markets, integrating technologies to safeguard sensitive data privacy, and efficiently leveraging distributed computing resources. Furthermore, the paper candidly examines two opposing viewpoints, acknowledging FL's current practical limitations while offering corresponding rebuttals and visionary considerations.

**Position:**

Yes

**Position In Title:**

Yes

**Related Work:**

2

**Strengths And Weaknesses:**

Strengths
1. The paper highlights the current controversies surrounding machine learning within the data and computing power supply chain, along with its monopolistic consolidation trends. Redefining federated learning from a traditional privacy tool into an alternative market mechanism offers a refreshing perspective for the field.
2. The paper proposes that FL transforms market transactions from “raw data” to “model updates,” thereby standardizing pricing and protecting small data owners from price discrimination.
3. Regarding counterarguments, the authors clearly acknowledge the vulnerabilities in their position.

Weaknesses*
1. The paper proposes using FL to address computational power and data monopolization, yet sidesteps the insurmountable engineering challenges at the foundational model scale. In cross-device scenarios, training LLMs faces severe constraints from communication bandwidth bottlenecks, memory and computational power ceilings on edge devices, and high system heterogeneity. The paper should provide a thorough system-level argument for the technical feasibility of this concept under the “scale law.”
2. The economic assumption regarding “model update replacement” is flawed. As discussed in Section 3.2, the authors naively assume that model updates can mask variations in the quality of original data, thereby enabling quality-agnostic equivalent  transactions. This is unreasonable. High-quality client data yields far greater gains for FL than low-quality data. Treating model updates as homogeneous, equivalent commodities is erroneous when confronting updates that are highly non-iid or even maliciously poisoned.
3. The computing power market proposal seems somewhat detached from reality. In Section 5.2, the author assumes that the price of model updates is underpinned by the fundamental value and cost of underlying computing power. However, FL's core value lies in leveraging private data to enhance global models. Therefore, its pricing should strictly depend on the “marginal utility” potential of client updates to the global model, rather than merely the consumption of computing power. Furthermore, the computing power of ordinary users' personal devices is extremely weak, and their data is highly fragmented, making them likely to be ignored by the market.
4. Although the paper introduces opposing viewpoints in Sections 6 and 7, its counterarguments appear insufficiently robust. Particularly in addressing the practical limitations of large-scale deployment within the FL framework, the paper fails to propose any convincing technical approach to overcome these challenges.

**Support:**

2

---

> ### Author Rebuttal · Authors · 2026-03-31
>
> We thank the reviewer for the engagement with our text and the utility of their comments for improving our manuscript. We will incorporate our responses below into the revised manuscript.
>
> W1: As this question is similar to other reviewers’ comments on computational capabilities, we direct the reviewer towards our rebuttal for Reviewer VwJp, points W2 and W4.
>
> W2: We would like to clarify our intended argument in Section 3.2. We fully agree with the reviewer that data quality has a large impact on how including specific data points or datasets in model training will affect the resulting model accuracy. Our point was that **data quality manifests in the model update(s) computed from the data,** as in FL the global model never sees the raw data, only the model updates coming from the clients. One can then assess the quality of the model updates directly, which may include not only data quality but also factors like computational capability, e.g., if a client uses smaller batch sizes due to limited compute or memory capabilities. The model updates themselves are certainly heterogeneous in general, and we will clarify this point.
>
> W3: Our intent in Section 5.2 was to elucidate the effects of FL deployments on the markets for computing resources. We agree with the reviewer that the “price” of FL model updates should include not just the value of the compute used to create the update but also the data on which the update is based (and its usefulness to the model being trained). Our intended point was that, while there is no single standardized method to measure the value of data to a given model, and in fact this value can vary depending on the other data points present and the specific model architecture in question, compute is a physical, fungible resource that thus lends itself to a more deterministic, physically backed valuation. We will further clarify this point.
>
> W4: We agree that FL poses novel technical challenges, and that these are important considerations when deploying it in practice. FL has, however, been deployed in practice, particularly in cross-silo settings where computing constraints are not as strict, suggesting that these technical challenges are not (always) insurmountable. Our goal in this paper was to draw attention to the larger economic benefits that may emerge from more widespread FL deployments and to encourage more research on (i) such connections to broader economic questions and (ii) technical solutions that can surmount practical challenges to FL deployments. Please see also our rebuttal for Reviewer VwJp, points W2 and W4, on the computational resources needed for FL and FL’s deployment in cross-silo settings.
>
> Q1: As this question is similar to other reviewers’ comments on computational capabilities, we direct the reviewer towards our rebuttal for Reviewer VwJp, points W2 and W4.
>
> Q2: As this question is similar to Reviewer VwJp’s comments regarding data poisoning and backdoors, we point the reviewer towards our rebuttal for Reviewer VwJp, point W1.
>
> Q3: Centralized aggregators may purchase these updates if that is the only legitimate way to access data for items sensitive to users such as personal data and browsing behavior. They could pay users for raw data, but this also introduces significant communication overhead due to the need to transfer this data to a central location. Moreover, AI model training is arguably entering a regime which is bottlenecked by computing demands. Paying users for raw data also retains security vulnerabilities from communicating sensitive information.
>
> Regarding the counterarguments to opposing viewpoints, we acknowledge their limitations given the paucity of space remaining for the latter sections. We would like to highlight research on lightweight implementations of FL such as low-rank finetuning [1] and smaller models. There is a wide surface of exploration in terms of making ML applications more agile and less computationally or data intensive. We will expand the relevant sections to better present our counterarguments.
>
> [1] A. Raje, et al. “Ravan: Multi-Head Low-Rank Adaptation for Federated Fine-Tuning”. NeurIPS, 2025.

---

> > ### Author Rebuttal · Reviewer_Kzjs · 2026-04-03
> >
> > Thank you for responding to my comments. I do not have any follow-up questions.

---

### Official Review · Reviewer_VwJp · 2026-03-12

**Significance:** 3
**Argument Clarity:** 3
**Rating:** 5
**Confidence:** 4

**Questions:**

1. Can use of server-less peer-to-peer system architecture address the limitation of FL described in Section 6.2?
2. What are current system-level challenges that prevent widespread use of FL? Is the application space of FL limited only due to current policies and practices or are there technical reasonings for this.
3. What kind of incentive framework could be expect to develop in FL, since you mentioned that it's fairly obscure when a server is coordinating the learning procedure.

**Alternative Views Section:**

Yes

**Compliance With Llm Reviewing Policy A Conservative:**

Affirmed.

**Discussion Potential:**

3

**Final Justification:**

The authors satisfactory addressed most of my concerns. I believe the paper is generally well-written and addresses a timely topic in a balanced fashion. It would indeed help the ML community to have a thoughtful discussion on the position presented in the paper. I would therefore like to increase my rating for the paper to "accept", and champion its acceptance.

**Paper Summary:**

The paper presents a clear position on the use of federated learning (FL) to tackle the growing scale of machine learning in terms of both compute and data. The paper states a well-argued position that FL can (i) alleviate concerns about proper consent and compensation for training data, (ii) protect privacy to some extent, and (iii) reduce pressure on the market for compute hardware by distributing the training procedure. Using FL as an example, the paper encourages research in system-level innovations to address rising conflicts in commercial machine learning concerning the use of private data and the monopoly of large tech firms. The paper also presents some counterpoints to their position, such as the limited application space of FL, its operational challenges, and the fact that FL cannot fully address the economic challenges (related to compensation for data) underlying modern machine learning applications.

**Position:**

Yes

**Position In Title:**

Yes

**Related Work:**

3

**Strengths And Weaknesses:**

**Strengths:**

1. The position is clear defined and argued for. At the same time, the paper presents counterpoints in detail.
2. The position is timely and well motivated by rising tension on data ownership and privacy regulations, and growing demand for computing resources.
3. While the position taken in the paper is often discussed amongst researchers working on FL, the paper also presents a somewhat novel perspective from economics viewpoint. The market-driven incentive for FL presented in the paper is very promising.
4. The paper has identified three key benefits of FL: fair data marker, data privacy and fair access to computing resources.

**Weaknesses:**

1. The paper has not discussed some of the technical challenges in FL, e.g., vulnerability to data and model poisoning [1], free-riders [2] and backdoors [3]. These vulnerabilities affect the quality and reliability of FL, and free-riding attacks pose a new challenge to data compensation.
2. The paper does not weigh the computational advantage of centralized ML with that of FL. FL has computational cost due to heterogeneity in data and system-level issues like asynchronicity. Moreover, the use of privacy mechanisms like homomorphic encryption and user-level differential privacy further increases the computational workload and learning error, respectively.
3. Privacy threats due to memorization are still valid in FL. In fact, at this point it is not clear whether FL can provide strong privacy.
4. Standard FL algorithms still require clients to download the large model to be updated. This already poses a big challenge in terms of computational resources needed at clients-level permitting participation of only big clients. This technical challenge should be discussed further.

**References:**
[1] Guerraoui, Rachid, Nirupam Gupta, and Rafael Pinot. "Byzantine machine learning: A primer." ACM Computing Surveys 56.7 (2024): 1-39.

[2] Nguyen, Thien Duc, et al. "{FLAME}: Taming backdoors in federated learning." 31st USENIX security symposium (USENIX Security 22). 2022.

[3] Fraboni, Yann, Richard Vidal, and Marco Lorenzi. "Free-rider attacks on model aggregation in federated learning." International conference on artificial intelligence and statistics. PMLR, 2021.

**Support:**

3

---

> ### Author Rebuttal · Authors · 2026-03-31
>
> We thank the reviewer for providing their thoughtful comments and discussion regarding FL. We would like to address the points in the review, which we will also incorporate into our manuscript:
>
> W1: Data and model poisoning, free-riding, and backdoors are indeed vulnerabilities in FL, and we agree that free-riding in particular touches on incentive challenges. We argue, however, that free riding can be alleviated with proper incentivization mechanisms, e.g., requiring a certain level of “useful” client participation in order to either receive payments or have access to the trained FL model. Much recent work in FL has additionally developed training frameworks that address poisoning and backdoor vulnerabilities, which may be incorporated into incentivization frameworks.
>
> W2: FL does indeed introduce computational costs by requiring local client training, and data heterogeneity across clients can slow down model convergence with respect to training rounds and/or wall-clock time, thus increasing the overall compute resources needed to train a model to a given accuracy level. Reducing this additional overhead is an active area of research, though we agree that some additional computational cost relative to centralized training is likely inevitable. However, FL may lead to higher utilization of available computing resources at clients, which may alleviate overall demand for compute resources. Investigating the tradeoffs between these drawbacks and the benefits of more democratized demand for compute power is an interesting research question and one we include in our call to action (see our response to Reviewer Jmdi, Q1).
>
> W3: We agree that vanilla FL does not provide ironclad privacy guarantees, although many FL variants address memorization attacks. However, centralized model training provides no privacy guarantees whatsoever, and compared to this starting point FL’s data locality may provide a promising start for ensuring privacy guarantees. For example, mechanisms like differential privacy can be incorporated into FL for stronger guarantees.
>
> W4: In practice, many currently-deployed FL systems are cross-silo scenarios (i.e., between institutions which have access to the resources for higher-performance training). Though not hyperscalers, these disparate institutions can nonetheless productively contribute to a model. Even in consumer-facing applications, cross-silo FL can be achieved as we look towards other federated structures such as social media (e.g., Mastodon and Bluesky federate users through local aggregation points that can act as clients) which can be composed of smaller companies and cooperatives to which users subscribe within a larger network of providers. Such companies can be assumed to have enough compute power to train large models.
>
> Q1: Peer-to-peer FL architectures can indeed alleviate some of the market power effects introduced by the presence of a central coordinator in FL, and we will add this point to the manuscript. However, it would not address the potential concerns of data itself concentrating in the hands of a few owners, e.g., media conglomerates, which could allow them to dominate the FL training.
>
> Q2: Beyond the challenges of communication, heterogeneity, and cost of FL infrastructure (W2) discussed elsewhere, we identify the lack of common protocols and orchestration challenges among participating parties as barriers to FL adoption. In addition, [1] identifies a fractured regulatory landscape as an additional barrier to implementation. The development of common FL frameworks and communication protocols is an area we invite researchers to design.
>
> [1] K. Kuo, C. Yadav, V. Smith. “Research in Collaborative Learning Does Not
> Serve Cross-Silo Federated Learning in Practice”. 2025.
>
> Q3: Incentive mechanisms in FL are an active area of research, and there are several economically plausible mechanisms that have been proposed. The most straightforward would involve a central server setting prices for each client’s participation, based on its estimated data quality and computation capabilities. However, other models like auctions, in which clients bid for the right to participate in a given model’s training, are also plausible. Both of these come with economic and usability challenges. First, it is not clear how a central server would set prices, or how a user would construct auction bids, as the “value” of user data to model accuracy is difficult to quantify without either soliciting private dataset information from the user or actually training the model. It is also unclear how to combine the “value” of this data with the value of users’ computational capabilities to determine the correct price that should be offered to each user acting as a FL client. Since users have limited visibility into model training, the resulting prices for different users may also appear arbitrary, which may reduce price acceptance.

---

> > ### Author Rebuttal · Reviewer_VwJp · 2026-04-02
> >
> > Thank you for responding to my comments. I do not have any follow-up questions.

---

### Decision · Program_Chairs · 2026-04-30

**Decision:**

Accept (regular)

**Comment:**

This position paper provides a timely perspective by analyzing Federated Learning through an economic and antitrust lens. By framing a familiar technical framework as a potential solution to systemic market risks, the authors offer a compelling meta-level perspective on the field's trajectory.

During the review process, reviewers raised valid concerns regarding the practical feasibility of deploying FL at the scale of modern foundation models, given the severe computational limits of edge devices. The authors provided an effective rebuttal that successfully refined their scope to emphasize "cross-silo" FL—where institutions rather than individual phones collaborate—making the compute requirements highly realistic. Furthermore, they expanded their call to action to include actionable technical solutions like integrating Parameter-Efficient Fine-Tuning (PEFT/LoRA) with FL.

While one reviewer remained hesitant about the novelty of the technical claims, the consensus is that bridging ML systems with macroeconomic analysis is a valuable contribution. This paper is well-argued, addresses a critical community bottleneck, and will stimulate constructive debate at the conference.